# Age and life expectancy clocks based on machine learning analysis of mouse frailty

Michael B. Schultz[1,9], Alice E. Kane[1,2,9], Sarah J. Mitchell[3], Michael R. MacArthur [3], Elisa Warner [4], David S. Vogel[5], James R. Mitchell[3], Susan E. Howlett [6], Michael S. Bonkowski[1,7] & David A. Sinclair [1,8⊠]

The identification of genes and interventions that slow or reverse aging is hampered by the lack of non-invasive metrics that can predict the life expectancy of pre-clinical models. Frailty Indices (FIs) in mice are composite measures of health that are cost-effective and non-invasive, but whether they can accurately predict health and lifespan is not known. Here, mouse FIs are scored longitudinally until death and machine learning is employed to develop two clocks. A random forest regression is trained on FI components for chronological age to generate the FRIGHT (Frailty Inferred Geriatric Health Timeline) clock, a strong predictor of chronological age. A second model is trained on remaining lifespan to generate the AFRAID (Analysis of Frailty and Death) clock, which accurately predicts life expectancy and the efficacy of a lifespan-extending intervention up to a year in advance. Adoption of these clocks should accelerate the identification of longevity genes and aging interventions.

[1] Blavatnik Institute, Department of Genetics, Paul F. Glenn Center for Biology of Aging Research at Harvard Medical School, Boston, MA, USA. [2] Charles Perkins Centre, The University of Sydney, Sydney, NSW, Australia. [3] Department of Genetics and Complex Diseases, Harvard T.H. Chan School of Public Health, Boston, MA, USA. [4] Department of Computational Medicine & Bioinformatics, University of Michigan, Ann Arbor, MI, USA. [5] Voloridge Investment Management, LLC and VoLo Foundation, Jupiter, FL, USA. [6] Departments of Pharmacology and Medicine (Geriatric Medicine), Dalhousie University, Halifax, NS, Canada. [7] Department of Dermatology, The Feinberg School of Medicine, Northwestern University, Chicago, IL, USA. [8] Department of Pharmacology, School of Medical Sciences, The University of New South Wales, Sydney, NSW, Australia. [9] These authors contributed equally: Michael B. Schultz, Alice E. Kane. ⊠email: david_sinclair@hms.harvard.edu

Aging is a biological process that causes physical and physiological deficits over time, culminating in organ failure and death. For species that experience aging, which includes nearly all animals, its presentation is not uniform; individuals age at different rates and in different ways. Biological age is an increasingly utilized concept that aims to more accurately reflect aging in an individual than the conventional chronological age. Biological measures that accurately predict health and longevity would greatly expedite studies aimed at identifying genetic and pharmacological disease and aging interventions.

Any useful biometric or biomarker for biological age should track with chronological age and should serve as a better predictor of remaining longevity and other age-associated outcomes than does chronological age alone, even at an age when most of a population is still alive. In addition, its measurement should be non-invasive to allow for repeated measurements without altering the health or lifespan of the animal measured[1]. In humans, biometrics and biomarkers that meet at least some of these requirements include physiological measurements such as grip strength or gait[2,3], measures of the immune system[4,5], telomere length[6], advanced glycosylation end-products[7], levels of cellular senescence[8], and DNA methylation clocks[9]. DNA methylation clocks have been adapted for mice but unfortunately these clocks are currently expensive, time consuming, and require the extraction of blood or tissue.

Frailty index (FI) assessments in humans are strong predictors of mortality and morbidity, outperforming other measures of biological age including DNA methylation clocks[10,11]. FIs quantify the accumulation of up to 70 health-related deficits, including laboratory test results, symptoms, diseases, and standard measures such as activities of daily living[12,13]. The number of deficits an individual shows is divided by the number of items measured to give a number between 0 and 1, in which a higher number indicates a greater degree of frailty. The FI has been recently reverse-translated into an assessment tool for mice which includes 31 non-invasive items across a range of systems[14]. The mouse FI is strongly associated with chronological age[14,15], correlated with mortality and other age-related outcomes[16,17], and is sensitive to lifespan-altering interventions[18]. However, the power of the mouse FI to model biological age or predict life expectancy for an individual animal has not yet been explored.

In this study, we track frailty longitudinally in a cohort of aging male mice from 21 months of age until their natural deaths and employ machine learning algorithms to build two clocks: FRIGHT age, designed to model chronological age, and the AFRAID clock, which is modeled to predict life expectancy. FRIGHT age reflects apparent chronological age better than FI alone, while the AFRAID clock predicts life expectancy at multiple ages. These clocks are then tested for their predictive power on cohorts of mice treated with interventions known to extend healthspan or lifespan, enalapril and methionine restriction. They accurately predict increased healthspan and lifespan, demonstrating that an assessment of non-invasive biometrics in interventional studies can greatly accelerate the pace of discovery.

## Results

### Frailty correlates with and is predictive of age.
We measured FI scores (Supplementary Fig. 1) approximately every 6 weeks in a population of naturally aging male C57BL/6Nia mice ($n = 60$) until the end of their lives. These mice had a normal lifespan, with a median survival of 31 months and a maximum (90th percentile) of 36 months (Fig. 1a and Supplementary Fig. 2). As expected, FI scores increased with age from 21 to 36 months at the population level (Fig. 1b). At the individual level, frailty trajectories displayed

**Table 1 Correlation between survival and delta age at individual ages.**

| Age | n | Delta age: FI score, Fig. 1f $r^2$ (p value) | Delta age: FRIGHT age, Fig. 3g $r^2$ (p value) | Survival: AFRAID clock, Fig. 4g $r^2$ (p value) |
|---|---|---|---|---|
| 21.0 | 14 | 0.021 (0.623) | 0.010 (0.728) | 0.035 (0.519) |
| 22.5 | 12 | 0.039 (0.537) | 0.005 (0.831) | 0.015 (0.708) |
| 24.0 | 13 | 0.390 (0.023*) | 0.296 (0.054) | 0.447 (0.012*) |
| 25.5 | 18 | 0.142 (0.124) | 0.012 (0.659) | 0.121 (0.157) |
| 27.0 | 26 | 0.103 (0.109) | 0.004 (0.753) | 0.218 (0.016) |
| 28.5 | 23 | 0.109 (0.124) | 0.031 (0.422) | 0.191 (0.037*) |
| 30.0 | 20 | 0.062 (0.291) | 0.019 (0.567) | 0.363 (0.004*) |
| 31.5 | 15 | 0.003 (0.859) | 0.011 (0.706) | 0.229 (0.071) |
| 33.0 | 12 | 0.305 (0.063) | 0.276 (0.079) | 0.292 (0.069) |
| 34.5 | 7 | 0.686 (0.021*) | 0.661 (0.026*) | 0.653 (0.028*) |
| 36.0 | 5 | 0.881 (0.018*) | 0.396 (0.256) | 0.201 (0.448) |

These data are for the testing dataset, and delta age is determined by either FI score or FRIGHT age, or AFRAID clock. Correlation ($r^2$) determined by Pearson correlation coefficients. *$p < 0.05$.

significant variance, representative of the variability in how individuals experience aging even within a population of inbred animals (Fig. 1c). As FI score was well correlated with chronological age, we sought to determine the degree to which FI score could model chronological and biological age. We performed a linear regression on FI score for age with a training dataset and evaluated its accuracy on a testing dataset (Fig. 1d–e). FI score was able to predict chronological age with a median error of 1.8 months, a mean error of 1.9 months, and an $r^2$ value of 0.642 ($p = 3.4e^{-38}$). We hypothesized that the error may be representative of biological age, with healthier individuals having a predicted age younger than their true age. We calculated this difference between predicted age and true age, termed delta age, and used remaining time until death as our primary outcome to compare with. For some individual age groups (24, 34.5, and 36 months), delta age did indeed have a negative correlation with survival, with biologically younger mice (those with a negative delta age) living longer at each individual age than biologically older mice (those with a positive delta age) (Fig. 1f and Table 1). For other groups this correlation is a trend, and more power may detect an association (Table 1). This suggests that the FI score is able to detect variation in predicted chronological age for mice of the same actual age, and this may represent biological age.

### Individual frailty items vary in their correlation with age.
While a simple linear regression on overall frailty score was somewhat predictive of age, we hypothesized that by differentially weighting individual metrics, we could build a more predictive model, as has been done with various CpG sites to build methylation clocks[9]. To this end, we calculated the correlation between each individual FI item and chronological age (Table 2). Some parameters, such as tail stiffening, breathing rate/depth, gait disorders, hearing loss, kyphosis, and tremor, are strongly correlated ($r^2 > 0.35$, $p < 1e^{-30}$) with age (Fig. 2), while others show very weak or no correlation with age (Table 2 and Supplementary Fig. 3). The fact that some parameters were very well correlated and others poorly correlated suggested that by weighting items we could build an improved model for biological age prediction.

### Multivariate regressions of frailty items to predict age.
We compared FI score as a single variable and four types of multivariate linear regression models to predict chronological age: simple least-squares regression, elastic net regression, random forest regression, and the Klemera–Doubal biological age

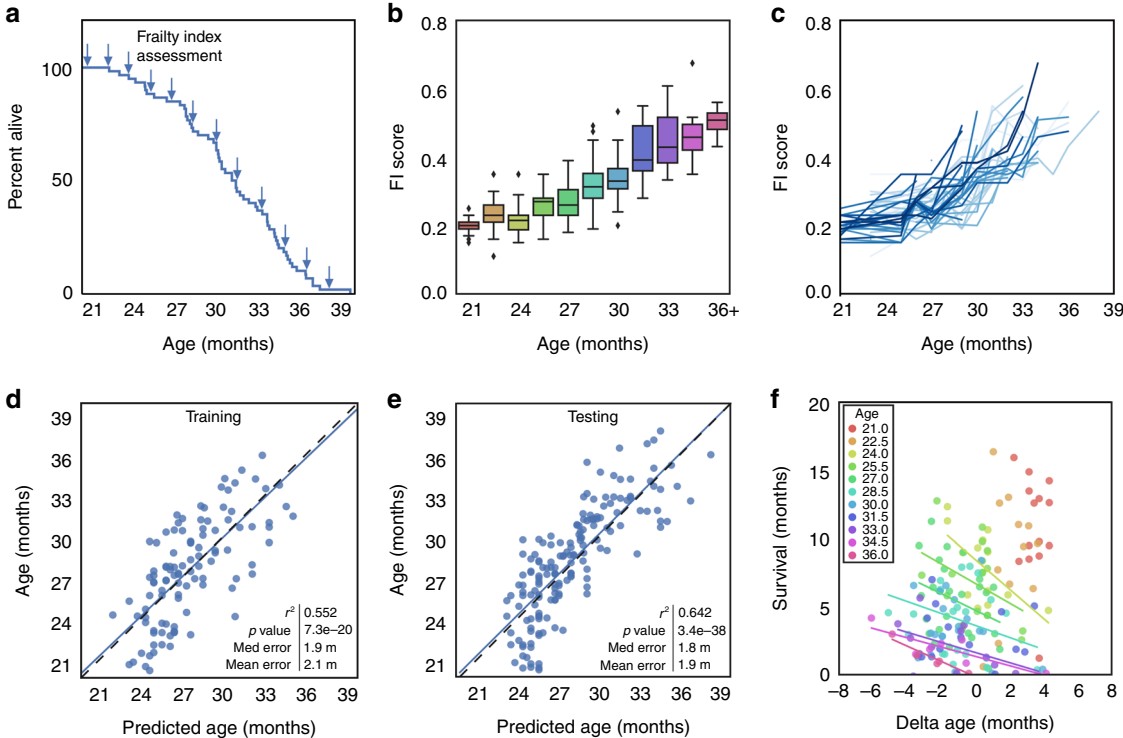

**Fig. 1 Frailty correlates with and is predictive of age in mice. a** Kaplan–Meier survival curve for male C57BL/6 mice ($n = 60$) assessed longitudinally for Frailty Index (FI) (indicated by arrows). **b** Box and whisker plots displaying median FI scores for mice from 21 to 36 months of age. Colors indicate different ages ($n = 24, 27, 20, 29, 43, 36, 32, 25, 18, 11, 6$). Box plots represent median, lower and upper quartiles, and 95 percentile. **c** FI score trajectories for each individual mouse from 21 months until death. **d** Univariate regression of FI score for chronological age on a training dataset, and **e** a testing dataset. For training and testing datasets, data were randomly divided 50:50, separated by mouse rather than by assessment, $n = 106$ datapoints for training and $n = 165$ for testing. Correlation determined by Pearson correlation coefficients. **f** Residuals of the regression (delta age), plotted against survival for individual ages (as demonstrated by different colors). Regression lines are only graphed for ages where there is an $r^2$ value >0.1. Source data are provided as a Source Data file.

estimation method (Eq. (1))[19]. We employed the bootstrap method on the training dataset to compare models. Only frailty items that had a significant, even if weak, correlation with age ($p < 0.05$) were included in the analysis (21 items, see Table 2). The multivariate models, particularly elastic net, the random forest, and the Klemera–Doubal methods (KDMs), were superior to FI as a single variable, with lower median error ($p < 0.0001$, $F = 49.46$, d.f. $= 499$) and mean error ($p < 0.0001$, $F = 68.37$, d.f. $= 499$, Supplementary Fig. 4a), higher $r^2$ values ($p < 0.0001$, $F = 57.1$, d.f. $= 499$), and smaller $p$ values ($p < 0.0001$, $F = 26.29$, d.f. $= 499$) when compared with one-way ANOVA. For further analysis, we selected the random forest regression model as it had the lowest median error (Fig. 3a–c). Random forest models can also represent complex interactions among variables, which linear regressions cannot do, and may perform better in datasets where the number of features approaches or exceeds the number of observations[20]. We term the outcome of this model FRIGHT age for *Fr*ailty *I*nferred *G*eriatric *H*ealth *T*imeline.

When assessed on the testing dataset, FRIGHT age had a strong correlation with chronological age, with a median error of 1.3 months, a mean error of 1.6 months, and an $r^2$ value of 0.748 ($p = 1.1e^{-50}$) (Fig. 3d, e). The items that were the largest contributors to FRIGHT age included breathing rate, tail stiffening, kyphosis, and total weight change (Fig. 3f). While FRIGHT age was superior to the FI score at predicting chronological age (Fig. 3a–c), the error from the predictions (delta age) were not well correlated with mortality (Fig. 3g). For the majority of individual age groups the $r^2$ values of the

correlation between FRIGHT age and survival were <0.1, indicating poor correlation (Table 1). Interestingly, the correlations were stronger for mice aged 34 months or greater, indicating that perhaps FRIGHT age is predictive of mortality only in the oldest mice (Table 1). This may be because the individual parameters that correlate well with chronological age are not necessarily the same as those that correlate well with mortality at all ages. Thus FRIGHT age has value as a predictor of apparent chronological age (e.g. this mouse looks 30 months old) but it is not yet clear whether it can serve as a predictor of other age-related outcomes.

**Multivariate regressions of frailty items to predict lifespan**. As FRIGHT age was not predictive of mortality at most ages, we sought to build a model based on individual FI items to better predict life expectancy. We began by calculating the correlation between each individual parameter and survival (number of days from date of FI assessment to date of death). Chronological age was the best predictor of mortality ($r^2 = 0.35$, $p = 1.9e^{-27}$), followed by FI score ($r^2 = 0.31$, $p = 2.7e^{-23}$), tremor, body condition score, and gait disorders (Table 3). However, many of these individual parameters appeared to be better predictors than they were, as a result of their covariance with chronological age. Their correlation with survival was largely only for mice of different ages, and not of the same age.

To build a model to predict mortality, we trained a regression using FI as a single variable, and multivariate regressions using the

**Table 2 Correlation coefficients ($r^2$) and $p$ values for frailty items with chronological age.**

| Item | $r^2$ | $p$ value |
|---|---|---|
| Tail stiffening | 0.58 | <0.001 |
| Breathing rate/depth | 0.50 | <0.001 |
| Gait disorders | 0.42 | <0.001 |
| Hearing loss | 0.38 | <0.001 |
| Kyphosis | 0.38 | <0.001 |
| Tremor | 0.38 | <0.001 |
| Body condition score | 0.26 | <0.001 |
| Forelimb grip strength | 0.20 | <0.001 |
| % twc | 0.12 | <0.001 |
| Menace reflex | 0.11 | <0.001 |
| Alopecia | 0.10 | <0.001 |
| Tumors | 0.08 | <0.001 |
| Diarrhea | 0.05 | <0.001 |
| Penile prolapse | 0.05 | <0.001 |
| Microphthalmia | 0.05 | <0.001 |
| Dermatitis | 0.05 | <0.001 |
| Rectal prolapse | 0.04 | <0.001 |
| Distended abdomen | 0.04 | <0.001 |
| Eye discharge/swelling | 0.04 | <0.001 |
| Coat condition | 0.04 | <0.001 |
| Body weight score | 0.02 | 0.01 |
| Threshold % rwc | 0.01 | 0.05 |
| Loss of fur color | 0.01 | 0.06 |
| Piloerection | 0.01 | 0.07 |
| Mouse grimace scale | 0.01 | 0.08 |
| Vestibular disturbance | 0.01 | 0.16 |
| Vision loss | 0.00 | 0.30 |
| Loss of whiskers | 0.00 | 0.33 |
| % rwc | 0.00 | 0.49 |
| Cataracts | 0.00 | 0.68 |
| Corneal capacity | 0.00 | 0.72 |
| Nasal discharge | 0.00 | 1.00 |

*Twc* total weight change, *rwc* recent weight change. Correlation ($r^2$) determined by Pearson correlation coefficients.

top and bottom quartiles, demonstrated a clear association with mortality risk for all age groups (Fig. 4h–k). These results suggest that the AFRAID clock may be useful for comparing the lifespan effects of interventional studies in mice many months before their death.

**Effect of interventions on FRIGHT age and AFRAID clock.** One ultimate utility for biological age models would be to serve as early biomarkers for the effects of interventional treatments, which are expected to extend or reduce healthspan and lifespan. A recently published study measured FI in 23-month-old male C57BL/6 mice treated with the angiotensin-converting enzyme (ACE) inhibitor enalapril ($n = 21$) from 16 months of age, or age-matched controls ($n = 13$)[21]. As previously published, enalapril reduced the average FI score compared to control-treated mice (Fig. 5a). When FRIGHT age was calculated for these mice, the enalapril-treated mice appeared to be a month younger than the control mice (control $27.8 \pm 1.1$ months; enalapril $26.8 \pm 1.4$ months, $p = 0.046$, $t = 2.1$, d.f. $= 32$) (Fig. 5b). When the data were converted to a prediction of survival with the AFRAID clock, the enalapril-treated mice were not predicted to live longer (control $5.9 \pm 0.7$ months; enalapril $6.2 \pm 0.9$ months, $p = 0.29$, $t = 1.09$, d.f. $= 32$) (Fig. 5c). This is interesting in light of the fact that enalapril has been shown to improve health, but not maximum lifespan, in mice[21,22].

Methionine restriction is a robust intervention that extends the healthspan and lifespan of C57Bl/6 mice[23–25]. We placed mice on a methionine restriction (0.1% methionine, $n = 13$) or control ($n = 11$) diet, from 21 months of age. We assessed frailty at 27 months of age and calculated FI, FRIGHT age and AFRAID clock. The methionine-restricted mice had significantly lower FI scores (control $0.37 \pm 0.30$; MR $0.30 \pm 0.04$, $p = 0.0009$, $t = 3.8$, d.f. $= 22$) (Fig. 5d), as well as a FRIGHT age 0.7 months younger than control-fed mice (control $29.8 \pm 0.9$ months; MR $29.1 \pm 0.6$ months, $p = 0.039$, $t = 2.19$, d.f. $= 22$) (Fig. 5e). Using the AFRAID clock, the methionine-restricted mice were predicted to live 1.3 months longer than controls (control $3.0 \pm 1.0$ months; enalapril $4.3 \pm 1.0$ months, $p = 0.006$, $t = 3.02$, d.f. $= 22$) (Fig. 5f). These analyses demonstrate that the FRIGHT age and AFRAID clock models are responsive to healthspan and lifespan-extending interventions.

## Discussion

This is the first study to measure the clinical FI longitudinally in a population of naturally aging mice that were tracked until their natural deaths in order to predict healthspan and lifespan. We show that the FI is not only correlated with but is also predictive of both age and survival in mice, and we have used components of the FI to generate two clocks: FRIGHT age, which models apparent chronological age better than the FI itself, and the AFRAID clock, which predicts life expectancy with greater accuracy than the FI. In essence, FRIGHT age is an estimation of how old a mouse appears to be, and the AFRAID clock is a prediction of how long a mouse has until it dies (a death clock). Finally, FRIGHT age and the AFRAID clock were shown to be sensitive to two healthspan or lifespan-increasing interventions: enalapril treatment and dietary methionine restriction.

The major advantage of the FI, and our models of the FI items, as aging biometrics is their ease of use. FI is quick and essentially free to assess, requires no specialized equipment or training, and has no negative impact on the health of the animals. We encourage future longevity studies to incorporate periodic frailty assessments as a routine measure into their protocols. This will help further determine the utility of frailty itself, as well as our FRIGHT age and AFRAID clock models, for predicting outcomes of interest, and may eventually be used as a screening tool to

FI items and chronological age with the simple least squares, elastic net, and random forest methods. All frailty items plus chronological age were included as variables in this analysis (32 items, see Table 3). As before, we compared these models using bootstrapping on the training set, and one-way ANOVA with Dunnett's post hoc test of $r^2$ value, $p$ value, median, and mean error (Fig. 4a–c and Supplementary Fig. 4). For prediction of survival, the elastic net and random forest regression models were the superior models, with higher $r^2$ values ($p < 0.0001$, $F = 36.62$, d.f. $= 399$), lower $p$ values ($p < 0.0001$, $F = 32.65$, d.f. $= 399$), and median errors ($p < 0.0001$, $F = 73.55$, d.f. $= 399$) than FI score alone (Fig. 4a–e and Supplementary Fig. 4). Similar results were obtained when chronological age was replaced with FRIGHT age, demonstrating that life expectancy can be accurately predicted with frailty measures alone (Supplementary Fig. 4c–f). We selected the random forest regression model (with chronological age) for further analysis, and we termed the outcome of this model the AFRAID clock for *Analysis of Frailty and Death*. The most important variables in the model were total weight loss, chronological age, and tremor, followed by distended abdomen, recent weight loss, and menace reflex (Fig. 4f). In the testing dataset, the AFRAID clock was well correlated with survival ($r^2 = 0.505$, median error $= 1.7$ months, mean error $= 2.3$ months, $p = 1.1e^{-26}$) (Fig. 4e). The AFRAID clock was also correlated with survival at individual ages (Fig. 4g) with $r^2 > 0.3$ and $p$ value <0.05 at 24, 30, and 34.5 months of age (Table 1). Plotting the survival curves of mice with the lowest and highest AFRAID clock scores at given ages, as determined by the

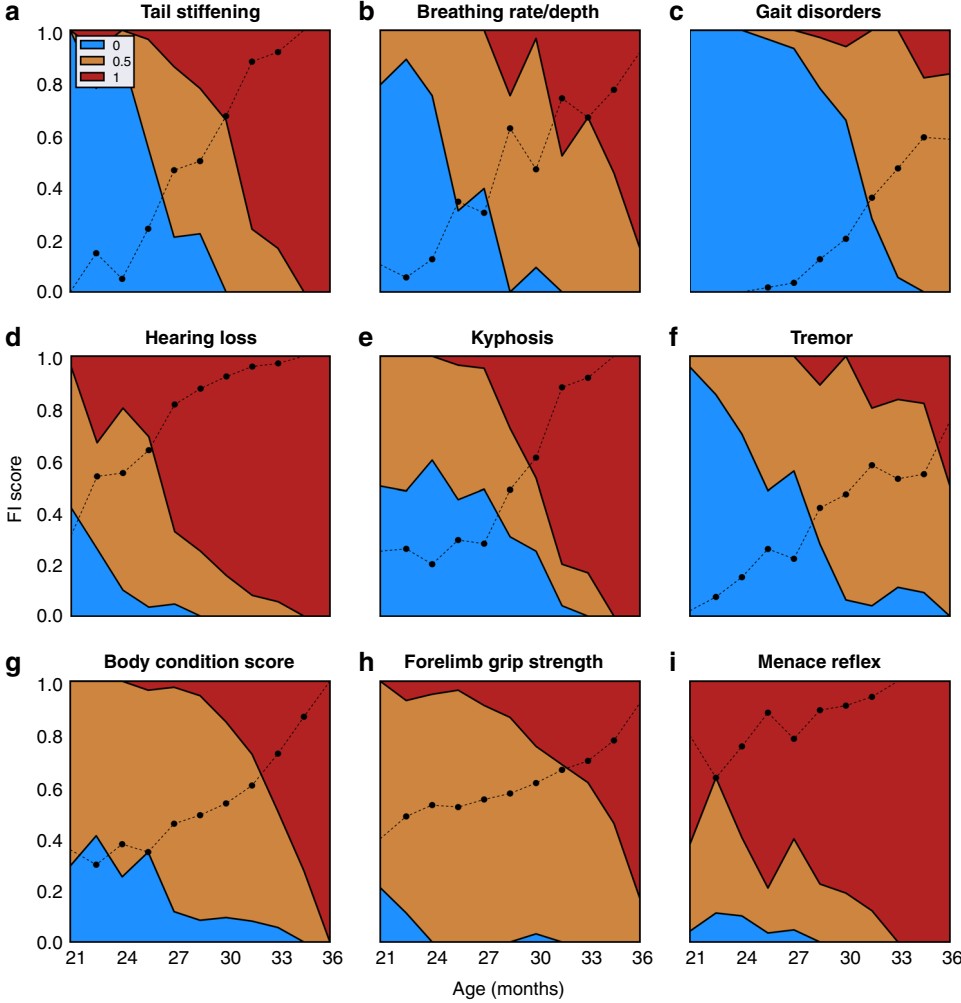

**Fig. 2 Individual FI items vary in their correlation with age.** Mean scores across all mice (black line) for the top nine individual items of the Frailty Index (FI) that were correlated with chronological age. Colors indicate proportion of mice at each age with each score (0, blue; 0.5, orange; 1, red). Source data are provided as a Source Data file.

decide whether to continue expensive interventional longevity studies after a short duration. Additionally, use of these non-invasive frailty measures in longevity studies will enable researchers to detect not only possible changes in lifespan, but also healthspan, arguably a more important outcome. We have created a website that automatically calculates and graphs FRIGHT age and AFRAID scores based on uploaded FI data, along with additional details of how to assses the frailty items in mice including a video demonstration (http://frailtyclocks.sinclairlab.org/) (Supplementary Fig. 6). Code for our clock calculators is also available on github (https://github.com/SinclairLab/frailty).

DNA methylation clocks are also promising biomarkers of biological age. In humans, these clocks are highly correlated with chronological age, and are able to predict, at the population level, mortality risk and risk of age-related diseases[11,26–31]. Methylation clocks have also been developed for mice, and shown to correlate with chronological age, and respond to lifespan-increasing interventions such as calorie restriction[32–35], but their association with mortality has not yet been explored. However, the major drawback of these mouse clocks is that they require repeated invasive blood collections and time-consuming and expensive data acquisition and analysis procedures.

This is the first time, to our knowledge, that frailty has been used to predict individual life expectancy in either humans or mice. In mice, frailty has previously been associated with mortality[17,36] but not used to predict lifespan. Mortality measures in mice that have focused on prediction, have either concentrated on the acute prediction of death such as in the context of sepsis[37,38], focused on only a few measures resulting in low or moderate correlations with survival[39–44], or used short-lived mouse strains[5]. The AFRAID clock, which was modeled in the commonly used C57BL/6 mouse strain and includes 33 variables, is able to predict mortality with a median error of 53 days across multiple ages. The real value of a biological age measure for mice, however, is in predicting how long individual mice of the same chronological age will live. The AFRAID clock was also able to predict mortality at specific ages, even as early as 24 months (approximately 6 months before the average lifespan, and 12 months before maximum lifespan without intervention). Additionally, when chronological age was replaced by FRIGHT age (predicted chronological age) to build a survival model similar to the AFRAID clock, we saw a similar accuracy of life-span prediction (Supplementary Fig. 4), indicating that life expectancy can be accurately predicted from FI items alone, without using chronological age as a variable.

This ability to predict expected lifespan in mice of the same chronological age provides exciting evidence that the AFRAID clock could be used in interventional longevity studies to understand whether an intervention is working to delay aging at

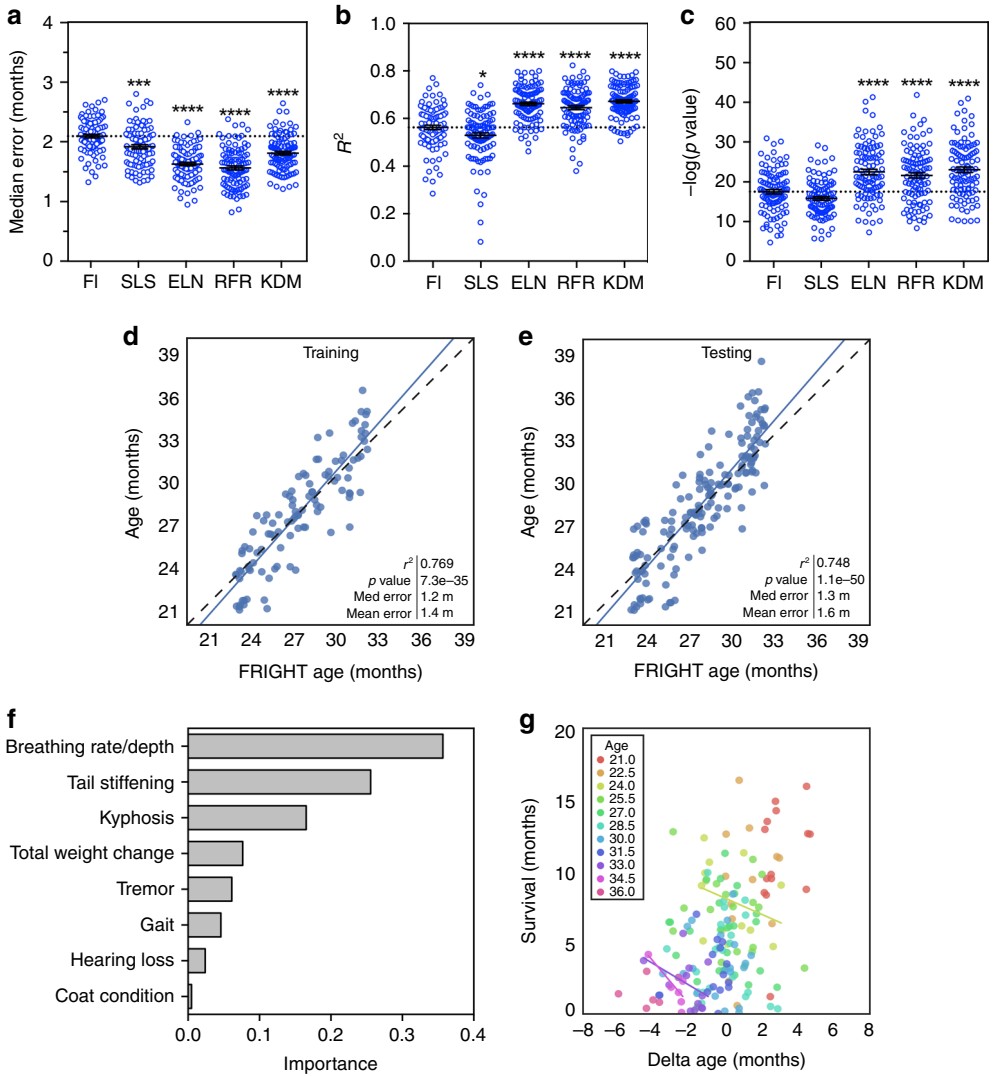

**Fig. 3 Multivariate regressions of individual FI items to predict age (FRIGHT age). a–c** Median error, $r^2$ values and $p$ values for univariate regression of Frailty Index (FI) score, and multivariate regressions of the individual FI items using either simple least squares (SLS), elastic net (ELN), the Klemera–Doubal method (KDM), or random forest regression (RFR) for chronological age in the mouse training set. All models were tested with bootstrapping with replacement repeated 100 times, and each bootstrapping incidence is plotted as a separate point. ****$p < 0.0001$ and ***$p < 0.001$ compared to FI model with one-way ANOVA. Error bars represent standard error of the mean. **d, e** Random forest regression of the individual FI items for chronological age on training and testing datasets (data was randomly divided 50:50, separated by mouse rather than by assessment, $n = 106$ datapoints for training and $n = 165$ for testing). This model is termed FRIGHT (Frailty Inferred Geriatric Health Timeline) age. Correlation determined by Pearson correlation coefficients. **f** Importances of top eight items included in the FRIGHT age model. **g** Residuals of the regression (delta age) plotted against survival for individual ages (as demonstrated by different colors). Regression lines are only graphed for ages where there is an $r^2$ value >0.1. Source data are provided as a Source Data file.

an earlier time point than death. Indeed, we show in the current study that treatment with the ACE inhibitor enalapril reduced FRIGHT age compared to controls but did not change the AFRAID clock. Enalapril is known to increase healthspan but not lifespan[22], indicating the value of these measures in detecting healthspan improvements even in the absence of an increase in lifespan. The dietary intervention of methionine restriction is known to increase healthspan and lifespan[23–25], and we saw reduced FRIGHT age and increased AFRAID clock scores in methionine-restricted mice at 27 months compared to controls. This means that had this been a longevity study, these measures would have given an indication of the lifespan outcomes less than halfway through the predicted study timeframe. In the methionine restriction experiment, the predicted age values for this independent cohort were slightly higher than their true values,

likely as a result of different baseline variability in frailty in different facilities. Similar effects have been seen with the mouse DNA methylation clocks[33,35]. Even so, there were still clear differences detected between groups, indicating both the importance of comparing results to controls within studies, and the ultility of these clocks even for independent mouse cohorts in different facilities.

Studies in humans have used the FI to determine increased risk of mortality within specific time periods[45–48], but not to predict individual life expectancies, as we have done here for mice. In theory the AFRAID clock could be easily adapted to predict mortality from human FI data. This has likely not been done as of yet, as it would require a large dataset that includes longitudinal assessments of FI items with mortality follow-up. This type of study is rare, particularly in an aging population. Even large

**Table 3 Correlation coefficients ($r^2$) and *p* values for frailty items with life expectancy.**

| Item | $r^2$ | *p* value |
|---|---|---|
| Age (days) | 0.35 | <0.001 |
| Tremor | 0.25 | <0.001 |
| Body condition score | 0.20 | <0.001 |
| Gait disorders | 0.19 | <0.001 |
| Tail stiffening | 0.19 | <0.001 |
| Breathing rate/depth | 0.17 | <0.001 |
| Hearing loss | 0.17 | <0.001 |
| Kyphosis | 0.12 | <0.001 |
| Distended abdomen | 0.11 | <0.001 |
| Menace reflex | 0.08 | <0.001 |
| % twc | 0.07 | <0.001 |
| Forelimb grip strength | 0.07 | <0.001 |
| Alopecia | 0.05 | <0.001 |
| Threshold % rwc | 0.04 | <0.001 |
| Microphthalmia | 0.03 | <0.001 |
| Body weight score | 0.03 | <0.001 |
| Coat condition | 0.03 | <0.001 |
| Diarrhea | 0.03 | 0.01 |
| Mouse grimace scale | 0.02 | 0.01 |
| Loss of fur color | 0.02 | 0.01 |
| Dermatitis | 0.02 | 0.03 |
| Piloerection | 0.02 | 0.03 |
| % rwc | 0.01 | 0.06 |
| Rectal prolapse | 0.01 | 0.07 |
| Penile prolapse | 0.01 | 0.11 |
| Tumors | 0.01 | 0.13 |
| Eye discharge swelling | 0.01 | 0.17 |
| Vestibular disturbance | 0.01 | 0.19 |
| Vision loss | 0.01 | 0.27 |
| Cataracts | 0.00 | 0.38 |
| Corneal capacity | 0.00 | 0.44 |
| Loss of whiskers | 0.00 | 0.51 |
| Nasal discharge | 0.00 | 1.00 |

*Twc* total weight change, *rwc* recent weight change. Correlation ($r^2$) determined by Pearson correlation coefficients.

cohort studies such as NHANES do not include enough people aged over 80 to allow for their specific ages to be released due to risk of identification. It would be interesting in future research to apply machine learning algorithms such as those used in the current study to predict individual life expectancy using FI data in humans.

We explored a range of regression techniques in the current paper. Simple linear and elastic net regressions are easily applied and interpreted, but are limited by being parametric and only considering linear relationships between variables, which reduce their predictive power for our data. The KDM, which was developed specifically to predict biological age by combining linear regressions of individual biomarkers[19], has been shown to predict human mortality risk[49,50]. Here, we applied this method to mice and saw some improved prediction over simple linear regression. For our final models, we used random forest algorithms, which are robust to outliers and noise, and allow for complex non-parametric modeling[20]. There are some limitations of these complex models, however, including a lack of interpretability of the weighting and interactions of the variables. Some previous studies have also used machine learning approaches for the development of aging biomarkers, including deep neural networks of standard blood biomarkers[51,52] and deep learning of brain imaging data[53], with promising results[54,55]. These have been exclusively humans studies, and our findings suggest that future studies exploring biological age biomarkers in mice could benefit from incorporating machine learning approaches such as neural networks or gradient boosting machine algorithms.

The aim of all three frailty metrics presented here, FI score, FRIGHT age, and the AFRAID clock, are robust methods for the appraisal of biological age. True biological age, however defined, is related to but separate from both chronological age and mortality, and without a clear biomarker with which to compare these three metrics, an assessment of their relative value is difficult. In one sense, FRIGHT age is the best because it tracks most closely with chronological age, with the variation in FRIGHT age (delta age; predicted−true age) representing biological age. An intervention that slows aging would likely suppress all aspects of aging including those that do not impact life expectancy (e.g. hair graying) and FRIGHT age would detect such changes. It is limited, however, by its lack of sensitivity in predicting mortality. In another sense, the AFRAID clock is the superior metric because an increase in life expectancy, median and maximum, is the current benchmark for the success of an aging intervention. One could also argue that overall unweighted FI is the best metric. While it is not best at predicting either chronological age or mortality, it is better than either FRIGHT age or AFRAID clock at predicting both. The best approach may be to employ all three estimates.

The predictive power of these models for both age and lifespan could be improved by the inclusion of larger *n* values (especially at the older ages), the assessment of frailty from ages younger than 21 months, and more complex modeling of the longitudinal aspects of our data. In the current study, we have used standard fixed-time predictive models treating each time point for each mouse as independent data, as there is currently no standard method for predicting outcomes at the level of the individual from data collected longitudinally[56,57]. Future studies could apply dynamic prediction approaches from the clinical biostatistics literature such as joint modeling[57,58] to develop models based on repeated measures of markers from the same mice. The models discussed in this study could also benefit from the incorporation of additional input variables, especially from relatively non-invasive molecular and physiological biomarkers or biometrics. Much can be inferred from tallying gross physiological deficits as has been done here with the mouse FI. These deficits, however, have cellular and molecular origins which may add predictive value at much earlier time points if they can be identified. FIs based on deficits in laboratory measures such as blood tests can detect health deficits before they are clinically apparent in both humans and mice[15,59]. Furthermore, this study used only male mice, and given the known sex differences in frailty, lifespan, and responses to aging interventions[15,60–62], it will be important to validate these models in female mice.

Ideal future studies will model biological age markers, not to predict chronological age or mortality alone, but rather a more complex composite measure of age-associated outcomes. Indeed, DNA methylation clocks that are trained on a surrogate biomarker and biometrics for mortality including blood markers and plasma proteins plus gender and chronological age[31,63] seem to have greater predictive power than those modeled on chronological age or mortality alone[64,65]. Future studies could develop a models based on the frailty items assessed here but modeled to predict a composite outcome including physiological measures in addition to chronological age. Still, even after the development of such composite clocks, the metrics described here—FI, FRIGHT age, and the AFRAID clock—will serve as rapid, non-invasive means to assess biological age and life expectancy, accelerating and augmenting studies to identify interventions that improve healthspan and lifespan.

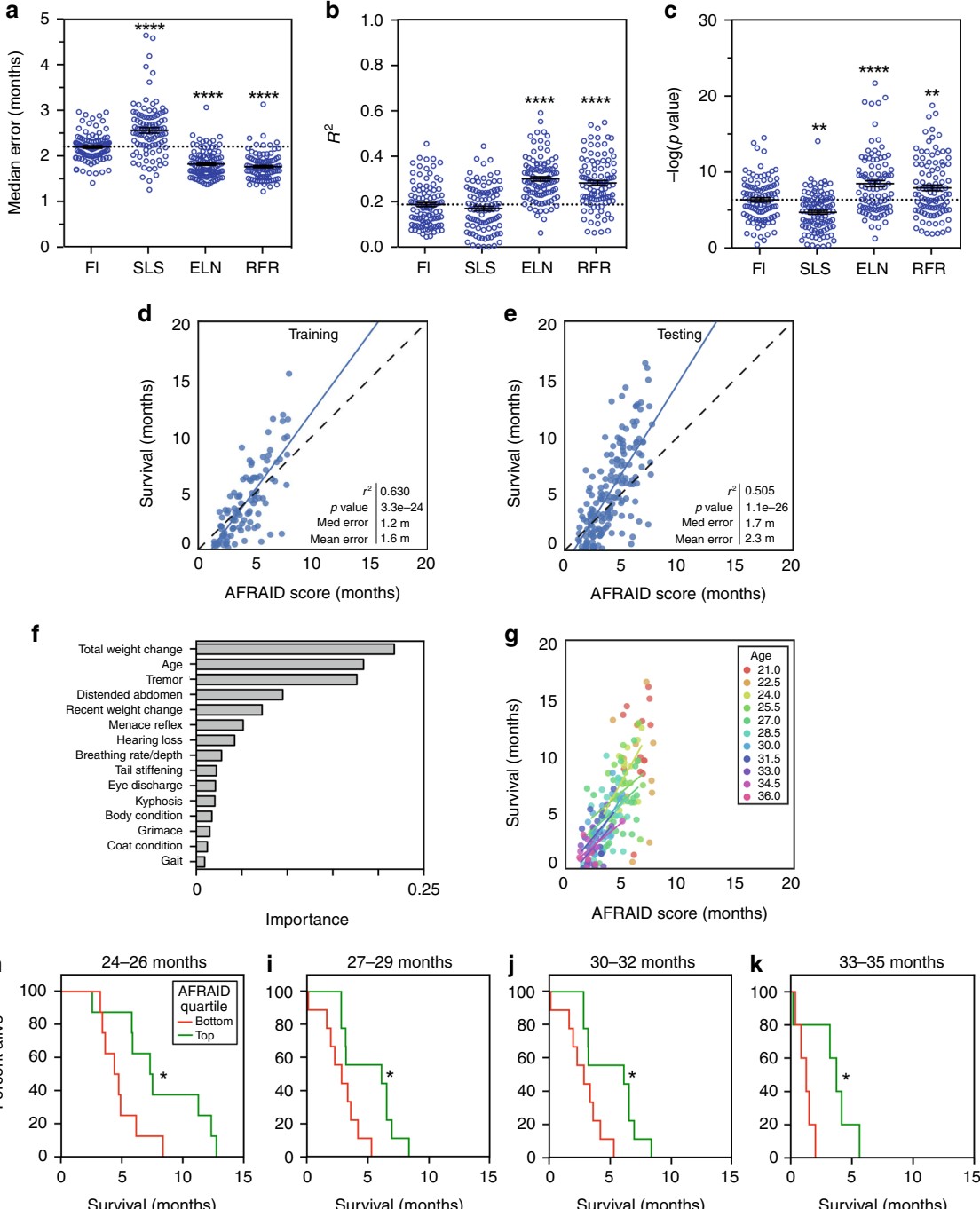

**Fig. 4 Multivariate regressions of FI items to predict life expectancy (AFRAID clock).** a–c Median error, $r^2$ values, and $p$ values for univariate regression of Frailty Index (FI) score, and multivariate regressions of the individual FI items using either simple least squares (SLS), elastic net (ELN), or random forest regression (RFR) for life expectancy in the mouse training set. All models were tested with bootstrapping with replacement repeated 100 times, and each bootstrapping incidence is plotted as a separate point. ****$p < 0.0001$ and ***$p < 0.001$ compared to FI model with one-way ANOVA. Error bars represent standard error of the mean. **d, e** Random forest regression of the individual FI items for life expectancy on training and testing datasets (data was randomly divided 50:50, separated by mouse rather than by assessment, $n = 106$ datapoints for training and $n = 165$ for testing), plotted against actual survival. This model is termed the AFRAID (*A*nalysis of *Frai*lty and *D*eath) clock. Correlation determined by Pearson correlation coefficients. **f** Importances of top 15 items included in the AFRAID clock. **g** AFRAID clock scores plotted against actual survival for individual mouse age groups (as demonstrated by different colors) in the testing dataset. Regression lines are only graphed for ages where there is an $r^2$ value >0.1. **h–k** Kaplan–Meier curves of the bottom (red lines) and top (green lines) quartiles of AFRAID clock scores for mice over 1–2 assessments at 24–26, 27–29, 30–32, and 33–35 months of age. *$p < 0.05$ compared with two-sided log-rank test. Exact $p$ values, respectively: 0.032, 0.015, 0.026, and 0.034. Source data are provided as a Source Data file.

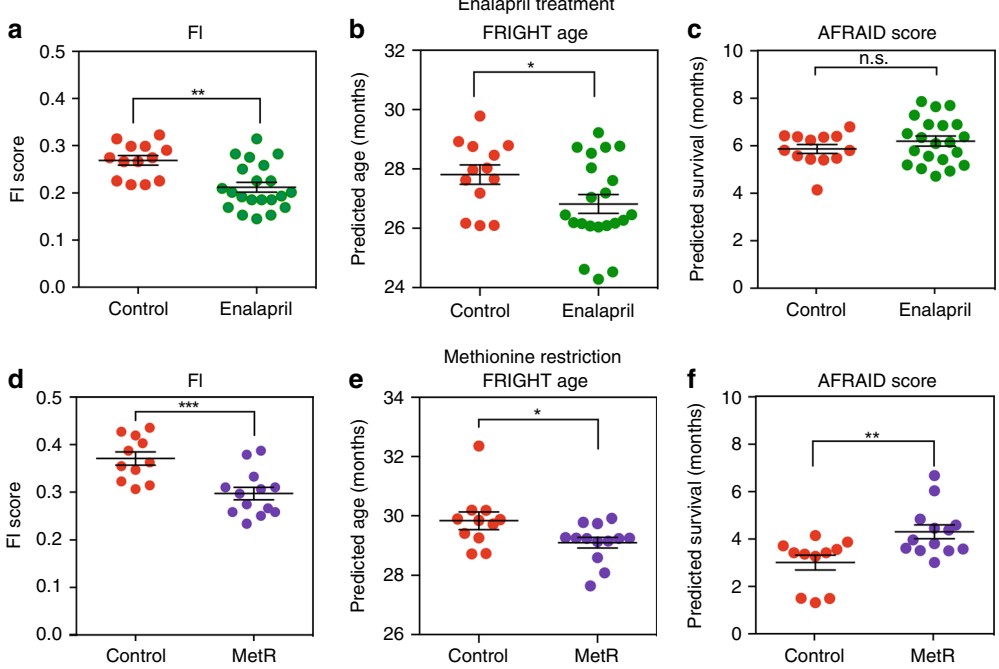

**Fig. 5 Response of FRIGHT age and AFRAID clock to interventions. a–c** Frailty Index (FI) score, FRIGHT (*Frailty Inferred Geriatric Health Timeline*) age and AFRAID (*Analysis of Frailty and Death*) clock for male 23-month-old C57BL/6 mice treated with enalapril-containing food (280 mg/kg) or control diet from 16 months of age. Data reanalyzed from previously published work[21]. Control n = 13, Enalapril n = 21. Exact *p* values, respectively: 0.001, 0.046, 0.29. **d–f** FI score, FRIGHT age, and AFRAID clock for male 27-month-old C57BL/6 mice treated with either a control diet (0.45% methionine) or methionine-restricted diet (0.1% methionine, MetR) from 21 months of age. *\*p* value <0.05, *\*\*p* < 0.01, and *\*\*\*p* < 0.001 compared with independent two-sided *t*-tests. Control n = 11, MetR n = 13. Exact *p* values, respectively, 0.001, 0.039, 0.006. Error bars represent standard error of the mean. Source data are provided as a Source Data file.

## Methods

**Animals**. All experiments were conducted according to the protocols approved by the Institutional Animal Care and Use Committee (Harvard Medical School). Aged males C57BL/6Nia mice were ordered from the National Institute on Aging (NIA, Bethesda, MD), and housed at Harvard Medical School in ventilated caging with a 12:12 light cycle, at 71 °F with 45–50% humidity. Mice were group housed (3–4 mice per cage) at the start of the experiment, although over the period of the experiment mice died and mice were left singly housed. A cohort of mice (n = 28) were injected with AAV vectors containing GFP as a control group for a separate longevity experiment at 21 months of age. This did not affect their frailty or longevity in comparison to the rest of the mice (n = 32), which were untreated (Supplementary Fig. 1). A total of 60 mice was used, which is consistent with other mouse longevity studies[66,67]. Both sets of animals had normal median (967 and 922 days) and 90th percentile (1078 and 1104 days) lifespans, slightly surpassing those cited by Jackson Labs (median 878 days, maximum 1200 days)[68,69], demonstrating that the mice were maintained and aged in healthy conditions. Mice were only euthanized if determined to be moribund (likely to die in the next 48 h) by an experienced researcher or a veterinarian based on exhibiting at least two of the following: inability to eat or drink, severe lethargy or persistent recumbence, severe balance or gait disturbance, rapid weight loss (>20% in one week), an ulcerated or bleeding tumor, and dyspnea or cyanosis. In these rare cases (n = 4, or 6.7%), the date of euthanasia was taken as the best estimate of death.

**Mouse frailty assessment**. Frailty was assessed longitudinally by the same researcher (A.E.K.), as modified from the original mouse clinical FI[14]. Malocclusions and body temperature were not assessed in the current study, so an FI of 29 total items was used. Individual FI parameters are listed in Supplementary Fig. 1. Briefly, mice were scored either 0, 0.5, or 1 for the degree of deficit they showed in each of these items with 0 representing no deficit, 0.5 representing a mild deficit, and 1 representing a severe deficit. For regression analyses, prediction variables were added to represent body weight change: total percent weight change, from 21 months of age; recent percent weight change, from 1 month before the assessment; and threshold recent weight change—mice received a score for this item if they gained more than 8% or lost more than 10% of their body weight from the previous month. For more details including images and video, see http://frailtyclocks.sinclairlab.org/. FI scoresheet for automated data entry (Supplementary Fig. 1g) is available online (https://github.com/SinclairLab/frailty).

**Intervention studies**. Data from enalapril-treated mice were reanalyzed from previously published work[21]. Briefly, male C57BL/6 mice purchased from Charles River mice were treated with control or enalapril food (30 mg/kg/day) from 16 months of age and assessed for the FI at 23 months of age.

*For the methionine restriction study*, male C57BL/6Nia mice were obtained from the NIA at 19 months of age and fed either a control diet (0.45% methionine) or methionine-restricted diet (0.1% methionine) from 21 months of age. Custom mouse diets were formulated at research diets (New Brunswick, NJ) (catalog #'s A17101101 and A19022001). Mice were assessed for the FI at 27 months of age.

**Modeling and statistics**. All analysis was done in Python version 3.6.x (jupyter (5.0.0), scikit-learn (0.19.0), pandas (0.20.1), numpy (1.14.0), scipy (1.0.0), seaborn (0.8.1)) or GraphPad Prism 6.0. Each time point of frailty assessment for each mouse is treated as independent. Training and testing datasets were randomly split 50:50 and were separated by mouse rather than by assessment resulting in n = 106 FI assessments (across 30 mice) for the training set and n = 165 assessments (across 30 mice) for the testing set. There were 7859 total datapoints included in the models, as calculated by 271 (106 + 165) assessments multiplied by 29 frailty items. Missing frailty data (18 individual datapoints out of 7859 total datapoints) were replaced by the median value for that item for that age group. Items included in the chronological age models were frailty assessment items with a significant (*p* < 0.05) correlation with age (21 items, Table 2). Items included in the lifepan models included all frailty items plus chronological age (32 items, Table 3). All models were assessed with bootstrapping with replacement, repeated 100 times. In each of those 100 iterations, the training set is divided into sub-training and validation sets, and the results on the validation sets are averaged over the 100 iterations. We held out the testing set for only reporting the final accuracy of the chosen model to prevent overfitting. The fit of the models was determined with the $r^2$ value which determines the proportion of the variance in our predicted outcome that is explained by the model, the median residual/error which represented the median difference between the actual and predicted outcome values, and the *p* value of the regressions. Median and mean error, $r^2$ and *p* values were compared across measures of FRIGHT age or AFRAID clock (Figs. 3a–c and 4a–c and Supplementary Fig. 4) with one-way ANOVA and Dunnett's post hoc test. Kaplan–Meier survival curves of the highest and lowest quartiles of AFRAID clock scores (Fig. 4) were compared with the log-rank test. FI, FRIGHT age, and AFRAID clock scores across intervention and control groups (Fig. 5) were compared with independent samples two-sided *t*-tests. For all statistics, *p* values less than 0.05 were considered significant. All data are presented as mean ± SD, except error bars on figures indicate standard error of the mean. For some graphs (Figs. 1d, e, 3d, e and Supplementary Fig. 2B), datapoints were jittered by up to ±0.5 months to improve data visualization.

Least squared and elastic net regressions were performed using algorithms provided in the Scikit-learn package[70] in Python. Least-squared regression was performed using the standard LinearRegression algorithm (copy_X = True; fit_intercept=True; n_jobs=None; normalize=False). Elastic net was performed with the ElasticNet algorithm with coefficients restrained as positive for FRIGHT age and negative for AFRAID score. Hyperparameters (FRIGHT: alpha = 0.2, l1_ratio = 0.9; AFRAID: alpha = 1.0, l1_ratio = 0.1) were chosen using bootstrapping. (All other hyperparameters were set to default: copy_X = True; fit_intercept=True; max_iter=100,000; normalize=False; precompute=False; selection=cyclic; tol=0.0001.) Standard, rather than survival analysis-oriented, versions of these regression algorithms were used as we have no censored data in our dataset, and we are treating our longitudinal datapoints as independent.

We calculated Klemera–Doubal biological age of each mouse using the methods first described by Klemera and Doubal[19] and later demonstrated by Levine[49] and Belsky et al.[71]. The KDM uses multiple linear regression but improves upon this by reducing multicollinearity between biological variables, which are intrinsically correlated. The KDM method consists of $m$ regressions of age against each of $m$ predictors. A basic biological age is then predicted based on the following equation (1):

$$BA_E = \frac{\sum_{j=1}^{m}(x_j - q_j)\left(\frac{k_j}{s_j^2}\right)}{\sum_{j=1}^{m}\left(\frac{k_j}{s_j^2}\right)^2},$$

where $k_j$, $q_j$, and $s_j$ represent the slope, intercept, and root mean square error of each of the $m$ regressions, respectively. While Klemera and Doubal further suggest using chronological age as a corrective term to limit the bounds of each predicted value, we used the version of the algorithm without age as, for the purposes of this study, we wanted to demonstrate the utility of the variables alone as predictors of age without knowledge of the true chronological age of the mouse.

Random forests are a type of machine learning algorithm which combines many decision trees into one regression outcome[20]. Compared to least squared and elastic net regressions, random forests have the advantage of being non-parametric and detecting non-linear relationships. Random forest modeling was performed using the Scikit-learn RandomForestRegressor algorithm[70]. Models were made with 1000 trees, and the minimum number of samples required for a branch split was limited to prevent overfitting as determined through bootstrapping (FRIGHT: min_samples_leaf=9; AFRAID: min_samples_leaf=6). (All other parameters were set to default: bootstrap=True; criterion=mse; max_depth=None; max_features=auto; max_leaf_nodes=None; min_impurity_decrease=0.0; min_impurity_split=None; min_samples_split=2; min_weight_fraction_leaf=0.0; n_jobs=None; oob_score=False.) We also computed and plotted the feature importance for each of the items with the highest value for this outcome. Feature importance is the amount the error of the model increases when this item is excluded from the model. Two example trees are shown in Supplementary Fig. 5.

**Reporting summary**. Further information on research design is available in the Nature Research Reporting Summary linked to this article.

## Data availability

The source data underlying all figures are provided as a Source Data File. Data are available at https://github.com/SinclairLab/frailty. Any remaining data supporting the findings of the study will be available from the corresponding author upon reasonable request

## Code availability

Code is available at https://github.com/SinclairLab/frailty.

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

## Acknowledgements

We would like to thank Alexander Colville, Doyle Lokitiyakul, and Yiming Cai for their help in carrying out the longevity study, and Maeve MacNamara and Daniel Vera for their help in setting up the website. This work was supported by the Glenn Foundation for Medical Research and grants from the NIH (R37 AG028730, R01 AG019719, R01 DK100263, R01 DK090629-08), and Epigenetics Seed Grant (601139_2018) from Department of Genetics, Harvard Medical School. A.E.K. is supported by an NHMRC CJ Martin biomedical fellowship (GNT1122542). Grants to S.E.H. from the Canadian Institutes for Health Research (PGT 162462) and the Heart and Stroke Foundation of Canada (G-19-0026260). E.W. is supported by an NIH Grant (5T32GM070449). Grant to J.R.M. from the NIH (2R56AG036712-06A1).

## Author contributions

M.B.S. and A.E.K. did most data analysis and wrote the manuscript; M.B.S. and M.S. B. conceived of and implemented the mouse lifespan study; A.E.K. did the frailty assessments; E.W. did the Klemera–Doubal analysis; S.J.M., M.R.M., and J.R.M. provided the methionine restriction data; S.E.H. provided the enalapril treatment data; D.S.V., J.R.M., S.J.M., M.R.M., and D.A.S. provided insight on study design and analysis; all authors read and edited the manuscript.

## Competing interests

D.A.S. is a founder, equity owner, advisor to, director of, consultant to, investor in and/or inventor on patents licensed to Vium, Jupiter Orphan Therapeutics, Cohbar, Galilei Biosciences, GlaxoSmithKline, OvaScience, EMD Millipore, Wellomics, Inside Tracker, Caudalie, Bayer Crop Science, Longwood Fund, Zymo Research, Immetas, and EdenRoc Sciences (and affiliates Arc-Bio, Dovetail Genomics, Claret Bioscience, Revere Biosensors, UpRNA and MetroBiotech, Liberty Biosecurity); Life Biosciences (and affiliates Selphagy, Senolytic Therapeutics, Spotlight Biosciences, Animal Biosciences, Iduna, Continuum Biosciences, Jumpstart Fertility (an NAD booster company), and Lua Communications); Iduna is a cellular reprogramming company, partially owned by Life Biosciences. D.S.V. sits on the board of directors of both companies. D.A.S. is an inventor on a patent application filed by Mayo Clinic and Harvard Medical School that has been licensed to Elysium Health; his personal share is directed to the Sinclair lab. For more information see https://genetics.med.harvard.edu/sinclair-test/people/sinclair-other.php. M.S.B. is a stockholder for MetroBiotech and Animal Biosciences, a division of Lifebiosciences. Other authors have no conflicts to declare.
