## [Peer Review File · Nature Communications]

Comments first round –

Reviewer #2 (Remarks to the Author):

Research in aging and age-related diseases is getting more and more important in times of an “aging” world population. Various interventions to slow progression of aging or address age-related diseases have recently been explored. In the first wave mainly lifespan extending drugs were explored, whereas focus has now shifted towards interventions increasing healthspan. Preclinical and clinical testing of such treatments requires reliable endpoints or biomarkers that predict health and life expectancy. One of such measures proposed is the FI, a list of non-invasive measures, based on the assumption that aging can be defined as accumulation of deficits. In the current manuscript FI and individual FI components from a longitudinal mouse aging study were used to generate mathematical models for chronologic aging (i.e. FRIGHT clock) and life expectancy (i.e. AFRAID clock) which superiority over just FI or chronologic age. In addition, these models/clocks were successfully applied to interventions known to affect health (i.e. enalapril) and lifespan (i.e. methionin restriction).

The manuscript addresses highly relevant questions in aging research. It is a nice piece of work to longitudinally assess FI and FI components to get to biomarkers. Such Biomarkers, in particular perspective markers for life expectancy, would be wonderful tools. Even though the authors attempt to explain the methods used it is not entirely clear what FRIGHT, AFRAID stands for, which FI components are included, how they are weighted or in exactly what dimensions “clock” have (see also Major Comments). To include these measures in future studies more details on how to determine them are needed. Another point needing some attention is “survival”. The methods are stating conditions when animals were euthanized and otherwise spontaneous death must have been the endpoint. It would be interesting to state how many animals were euthanized compared to natural death. This seems important as some of the euthanization criteria are overlapping with FI components. It is also a bit unfortunate that the intervention studies did not have survival as endpoint as this would have been the ultimate prove that AFRAID predicts survival accurately.

Major comments

Data and modelling

As mentioned above the mathematical and machine learning methods are not described in very detail. In order to use these clocks in other labs and studies more details are essential. It is also not totally clear if FRIGHT stands indeed for “chronologic” or rather “biologic” age. This needs a bit more clarification, in particular as the interventions reduce FRIGHT.

What is also quite astonishing is that the FRIGHT age predicted in the intervention studies is much higher than the actual age of the animals used (23 or 27 months at the end of the study). If looking at Fig. 3e this does not seem to be a general observation even though there is a trend at older age.

Figures/Tables: the labelling of the figures, Tables and text is not always consistent and makes it quite difficult to follow this complicated mathematic topic. For example, Table 1 shows correlations such as Delta Age – FRIGHT Age and refers to Fig. 3e. However, Fig. 3 e shows age and predicted age. It is highly recommended to check all this kind of references and also use “common” terminology throughout the manuscript.

Minor comments

Figure 1b: it would be great to add the number of animals used per age for the FI measurements. It seems that only 3 animals lived beyond 33 months. Maybe one could mark the animals which euthanized in Fig. 1 c.

Fig. 2: it is quite obvious that some deficits occur earlier (e.g. forelimb grip strength) and other much later (e.g. gait disorder). Still “gait disorder” seems to highly correlate with both chronologic age and life expectancy (Table 2/3, Fig 2). As the FI was only measured from 21 months onwards could some of the correlations been biased “towards” very old animals. If so, would this influence the modelling?

Reviewer #3 (Remarks to the Author):

The Manuscript presented by Schultz and Kane et al., takes a large and logical step forward in the field of frailty assessment in preclinical models by pairing the clinical frailty index with machine learning strategies. This effort will likely optimize the accuracy of prediction and better identify the contributions of each parameter. Additionally, this work advances novel approaches to examine biological age versus chronological age, which may possibly be translated from these preclinical models for human study. There were no major concerns in the review of this manuscript. There are some minor considerations:

1) Figure 3 and 4, panels D and E -- an N of 51 mice were split for training and testing subsets, however; the panel includes roughly 100 data points -- I assume this includes the same mice at different ages, in which case, perhaps this should be explained in the legend or the ages identified as in figure 3G.

2) The Random forest model was chosen for having a lower median error than the other approaches, although this difference was not statistically significant. Therefore, would it make sense to submit all the models that are statistically similar for training, and then subsequent testing to identify which provides the strongest correlation?

3) The authors indicate this is the first study to, "measure frailty longitudinally in a population of naturally aging mice that were tracked until their natural deaths in order to predict health span and lifespan." Although the distinction of the goal of predicting healthspan and lifespan is acknowledged, many of these concepts were examined recently using physical frailty assessment tools (Baumann et al., Aging 2018 PubmedID: 30562163)

4) Although it is a supplement, an additional description of what is being observed in figure S1 a-f would aid readability and understanding.

Reviewer #4 (Remarks to the Author):

The main claim of the paper regarding originality seems to be that it is 'the first study to measure frailty longitudinally in a population of naturally aging 209 mice that were tracked until their natural deaths in order to predict healthspan and lifespan' (first sentence of the Discussion section).

In general, consistently with this claim, frailty index studies tend to be based on cross-sectional data, but there are a few studies on frailty index using longitudinal data, involving some kind of prediction (at least of mortality); so, to be more precise, it should be briefly clarified the difference between this work and other works on frailty index using longitudinal data. Two such works are (Hoogendijk et al. 2017) (reference below) and (Rockwood et al. 2017) (cited in the paper), where the latter also measured frailty longitudinally in mice. Both these works seem to use the frailty index to predict mortality (see e.g. Figure 4 in (Rockwood et al. 2017)). One difference between the authors' work and those works seems to be that the authors are using several statistical and supervised machine learning techniques for prediction, whilst the above two works seem to have less focus on prediction and they focus on using classical statistical techniques for survival analysis (Kaplan-Meier curves, Cox regression).

Reference:

Hoogendijk, E. O., Theou, O., Rockwood, K., Onwuteaka-Philipsen, B. D., Deeg, D. J., & Huisman, M. (2017). Development and validation of a frailty index in the Longitudinal Aging Study Amsterdam. *Aging clinical and experimental research*, 29(5), 927-933.

In any case, I consider that this current work has enough originality for publication and will be of interest to others in the community, given the great importance of research on ageing biomarkers and frailty index research.

The set of regression techniques used seems appropriate overall, but there is not much justification for their choice. The way I see it, they represent a good diversity, varying from simple linear regression, with the advantage of easy interpretability but the disadvantage / limitation of

being strongly parametric and considering only linear relationships between the predictive variables and the target variable (see also another related comment below), to more powerful random forests, with the advantage of being non-parametric and detecting non-linear relationships but the disadvantage of being more difficult to interpret. Actually, it is not practical to interpret all trees in the forest, although there are kind of indirect interpretations based on ranking the variables in decreasing order of importance in the model, as used in the paper. I think it would be good to have a short discussion about these issues in the paper.

This work uses standard versions of regression algorithms (least squared, elastic net and random forests). There are versions of these algorithms developed specifically for survival analysis (e.g. random survival forests), but I assume these survival analysis-oriented versions were not used for two reasons: (a) although predicting survival time, there is no censored data, since all animals were followed until death; and in the absence of censored data the survival problem can be cast (although not perfectly) as standard regression; (b) each run of a regression algorithm is basically considering variables at a single time point, with a few exceptions (this is discussed in more detail below). This kind of rationale could be briefly mentioned in the Methods section.

In any case, standard regression methods like least square assume the target variable is normally distributed, which is unlikely to be the case for survival times. By contrast, random forests, as a non-parametric method, make no such assumption. This kind of issue should be briefly discussed in the Methods section.

I also think the way the longitudinal information was used could be clearer. If I understand it correctly, although the data is longitudinal in nature, each run of a standard (non-longitudinal) regression technique for predicting mortality is just using the variables measured at a given point in time, the time point where the estimation of survival / mortality is done. For instance, when training a regression model for predicting mortality at age 24 months, regarding the individual frailty index items, only the predictive variables measured at age 24 are considered, predictive variables at earlier or later ages are ignored. Temporal information is considered by the regression algorithms predicting mortality only in the case of a few variables explicitly encoding temporal information, namely total percent weight change, recent percent weight change and threshold recent weight change. This issue should be clearer.

(If this understanding is not correct, that would suggest an even stronger need to explain more clearly exactly how the longitudinal data was used, since other readers may get confused too.)

In line 385 it is said there are in total 2,460 'data points'. What exactly is a 'data point' in this context? How exactly was this total number calculated?

Several relevant related works on aging clocks are discussed, but considering that the title of the paper refers to machine learning, and considering that the state of the art in terms of predictive performance in machine learning is often deep learning (although with some limitations, see below), it seems reasonable that the discussion should be extended to mention also related work using deep learning for developing ageing biomarkers or estimating biological ageing. Such works are reviewed e.g. in:

Gialluisi et al. Machine learning approaches for the estimation of biological aging: the road ahead for population studies. *Frontiers in Medicine*, 6, Article 146, July 2019

Zhavoronkov et al. Deep biomarkers of aging and longevity: from research to applications. *Aging* 2019, 11(22), 10771-10780, www.aging-us.com

Regarding the limitations of deep learning methods, normally they require a very large amount of data, which would seem to prevent their use in this current work, and they also produce models which are very hard to interpret, so I think it is justifiable not to use them in this particular work. However, the aging clocks produced by such deep learning methods should be briefly discussed as part of related work, as mentioned above.

In the experiments, FRIGHT was the best predictor of age and AFRAID the best predictor of mortality. I understand that, when training the AFRAID regression models, one of the predictive variables was chronological age. I wonder what would be the effect of replacing chronological age

with the biological age estimated by the FRIGHT clock, when training the AFRAID model. Would that increase or decrease the predictive performance (on the test set) of AFRAID? That would be one approach to combine FRIGHT and AFRAID, a step in the direction of a hybrid clock as suggested in lines 283-285 of the Discussion section, and a simple experiment to do.

In addition to reporting results in terms of r^2 and median error, it would be good to report results in terms of Mean Absolute Error (MAE) too. The median error is a good measure of central tendency, but it should be complemented by a measure which captures information about errors across all samples in the test set, like the MAE.

The last paragraph of the Discussion section is discussing some limitations of this work and future research directions, overall good, but I think it is important to remind the reader here that this study has the limitation of using only data from male mice, and perhaps cite here a couple of papers showing the importance of studying variations of frailty in males vs females.

Line 228 mentions the clock calculators were made available online, which is good, but to allow reproducibility of the results, the full dataset (with $n = 51$, all the 30 or so items used to calculate the FI, and the survival time for each mouse) should be made available online too; this is very important.

More specific comments are as follows.

The title of Table 1 should specify that these results were computed on the test set (assuming this was the case).

Lines 175 and 176: 'The AFRAID clock was also correlated with survival at individual ages (Figure 4g) with $r^2 > 0.1$ and p value < 0.05 at 24, 27, 28.5, 30 and 34.5.'

This sentence is relying purely on statistical significance ($p < 0.05$) to highlight the main correlation results, almost ignoring the magnitude of the r^2 value. Statistical significance does not imply practical significance – see e.g. the following two references for a discussion why statistical significance is over-rated in practice and the limitations of using only p -values to select the significant results to be highlighted:

Amrhein, V., Greenland, S., McShane, B. (and more than 800 signatories) Retire statistical significance. *Nature*, Vol. 567, 21 March 2019, pp. 305-307.

Ronald L. Wasserstein, Allen L. Schirm & Nicole A. Lazar (2019) Moving to a World Beyond " $p < 0.05$ ", *The American Statistician*, 73:sup1, 1-19, DOI: 10.1080/00031305.2019.1583913

Although the issue of which r^2 value is high enough to be considered a strong or at least reasonable degree of correlation is obviously subjective, it seems reasonable to say that an r^2 of around 0.2 is not high enough to be relevant in practice, regardless of p -value. Out of the 5 age values mentioned above, the r^2 for 27 and 28.5 months are 0.218 and 0.191, much lower than the r^2 for the other 3 above ages. I would suggest to say that there are 3 ages with $r^2 > 0.3$ and statistically significant results, mentioning explicitly the $r^2 = 0.653$ for age 34.5 and $r^2 = 0.447$ for age 24, rather than reporting all the above 5 age values with r^2 having $p < 0.05$.

Lines 245-247 state that the AFRAID clock predicted mortality with a median error of 46 days across multiple ages. I understand this is the median error on the test set. In figure 4(g), the median error of AFRAID on the test set is 1.7 months. Assuming a month = 30 days, 1.7 month gives a median error of 51 days, not 46 days as stated in line 247.

Line 99 mentions median error of 1.8 months, r -squared 0.642, and $p = 7.3e 20$. The first two of these three values are the same as in Figure 1(e), and indeed the whole sentence seems to be about performance on the testing set, but the third above value, $p = 7.3e 20$, was wrongly taken from Figure 1(d), referring to performance on the training set. In Figure 1(e), the value is $p = 3.4-38$.

Lines 322-323 (Methods – Animals), it is stated that in the rare cases where mice were euthanized, the date of euthanasia was taken as the best estimate of death. The phrase 'In these

rare cases' is vague. To be more precise, please give the exact number of such rare cases.

Lines 394-395 (Methods – Random Forest), it is stated that 'the minimum number of samples required for a branch split was limited...', which is vague. To be precise, please give the exact value of the threshold used as minimum number of samples for a branch split. Actually, all settings of hyper-parameters of the RF algorithm have to be given, so please give also the used value of the mtry parameter (the number of randomly sampled variables used as candidates for branch split at each tree node), as well the settings of any other hyper-parameters of the RF algorithm (and for the other regression algorithms used too).

Even if the authors are using default parameter settings, it is worth mentioning explicitly their exact values, because default values in the software implementing an algorithm can change with time, and so citing the precise hyper-parameter settings makes the paper more self-contained.

Minor changes:

Lines 99, 139, 616: replace r-squared by r^2 (using superscript, of course, which does not appear in this plain text web form)

Line 174, replace 1.1-26 by 1.1×10^{-26}

Line 305, replace expectency by expectancy

We thank the reviewers for their positive comments on our manuscript. Reviewer 2 said “The manuscript addresses highly relevant questions in aging research,” and “perspective markers for life expectancy would be wonderful tools.” Reviewer 3 said the manuscript “takes a large and logical step forward in the field of frailty assessment in preclinical models,” that “this effort will likely optimize the accuracy of prediction,” that “this work advances novel approaches to examine biological age versus chronological age,” and that “there were no major concerns.” And reviewer 4 said this study “will be of interest to others in the community, given the great importance of research on ageing biomarkers and frailty index research.”

We also thank the reviewers for their highly constructive feedback. We have addressed each of the comments below.

Since submission of the initial manuscript, we have also launched a website (frailtyclocks.sinclairlab.org) corresponding with this manuscript, which we hope will serve as a resource to the scientific community to more easily use the tools developed in this study.

Reviewer 2

General comments

1. Even though the authors attempt to explain the methods used it is not entirely clear what FRIGHT, AFRAID stands for, which FI components are included, how they are weighted or in exactly what dimensions “clock” have (see also Major Comments).

We apologize for not being clear in our descriptions. We have updated the manuscript to make it clearer for the reader, including clarification of what each of the models stands for and which items are included in the results and methods sections. FRIGHT age stands for Frailty Inferred Geriatric Health Timeline and is a model to predict chronological age. AFRAID stands for Analysis of Frailty and Death and is a model to predict remaining lifespan. The items included in FRIGHT age are those frailty assessment items with $p < 0.05$ correlation with age ($n=21$ items, Table 2). All frailty assessment items plus age were included in AFRAID score ($n=32$ items, Table 3). While the weighting and dimensions are complex to describe, as the specific interpretability of random forest models is limited, we present the importance score of the variables in each model (Figure 3F and 4F), which is a measure of the amount the error in predictions increases when this item is excluded from the model. This gives some idea of which FI components contribute most significantly to each of the models. Considerations of the limitations of interpretability of random forests have been added to the discussion section of our manuscript.

2. To include these measures in future studies more details on how to determine them are needed.

We thank the reviewer for the suggestion. To clarify the methods for the readers and make these measures and models as widely used as possible, we have built a website with video and images of how to complete the frailty assessment in mice <http://frailtyclocks.sinclairlab.org/>. The new website also allows for any user’s data to be easily uploaded. FRIGHT age and AFRAID

scores are then automatically calculated, graphed, and can be downloaded. We have put a link to this website in the discussion of the manuscript (Figure S6).

3. Another point needing some attention is “survival”. The methods are stating conditions when animals were euthanized and otherwise spontaneous death must have been the endpoint. It would be interesting to state how many animals were euthanized compared to natural death. This seems important as some of the euthanization criteria are overlapping with FI components.

We have updated the methods with the exact criteria for euthanasia and highlighted that only 4 out of 60 mice were euthanized. These mice are also referred to in new supplementary figures S2c and d.

4. It is also a bit unfortunate that the intervention studies did not have survival as endpoint as this would have been the ultimate prove that AFRAID predicts survival accurately.

We agree with the reviewer that this would have been the best proof, but we unfortunately don't have this data. We hope future studies will further validate the clocks for use in intervention studies.

Major comments

5. Data and modelling: As mentioned above the mathematical and machine learning methods are not described in very detail. In order to use these clocks in other labs and studies more details are essential.

Thank you for highlighting this. We have added the sub-parameters for each of the models into the methods section of the manuscript, and also provided a website where others can use our models to calculate FRIGHT and AFRAID scores for their own data:

<http://frailtyclocks.sinclairlab.org/>. Additionally, all the code for our models is available on github: <https://github.com/SinclairLab/frailty>.

6. It is also not totally clear if FRIGHT stands indeed for “chronologic” or rather “biologic” age. This needs a bit more clarification, in particular as the interventions reduce FRIGHT.

FRIGHT is trained on chronological age, but the variation (delta age) may represent biological age. We have added further discussion of this to the manuscript.

7. What is also quite astonishing is that the FRIGHT age predicted in the intervention studies is much higher than the actual age of the animals used (23 or 27 months at the end of the study). If looking at Fig. 3e this does not seem to be a general observation even though there is a trend at older age.

This is an astute observation. We have now added a discussion of this limitation of that particular model. Mice in different facilities likely have different baseline variability in frailty variables, resulting in the higher predictions of age in these cohorts. However, the most

important comparison is between intervention and control groups within cohorts, and we are seeing expected differences in FRIGHT age and AFRAID scores in these cohorts. This has also been seen with other clocks, including mouse DNA methylation clocks, where the absolute ages may be different between cohorts, but the difference between groups within a study is still very meaningful.

8. Figures/Tables: the labelling of the figures, Tables and text is not always consistent and makes it quite difficult to follow this complicated mathematic topic. For example, Table 1 shows correlations such as Delta Age – FRIGHT Age and refers to Fig. 3e. However, Fig. 3e shows age and predicted age. It is highly recommended to check all this kind of references and also use “common” terminology throughout the manuscript.

Thank you for pointing this out. We have made all terminology consistent.

Minor comments

9. Figure 1b: it would be great to add the number of animals used per age for the FI measurements. It seems that only 3 animals lived beyond 33 months. Maybe one could mark the animals which euthanized in Fig. 1 c.

Good suggestion. We have added two new supplementary figures based on Figure 1a and 1c (S2c and d) that indicate how many animals were assessed at each timepoint and indicate the 4 animals that were euthanized. We also corrected the X-axis on Figure 1a to reflect the fact that we had 16 animals live beyond 33 months, with the oldest living to 39.7 months of age.

10. Fig. 2: it is quite obvious that some deficits occur earlier (e.g. forelimb grip strength) and other much later (e.g. gait disorder). Still “gait disorder” seems to highly correlate with both chronologic age and life expectancy (Table 2/3, Fig 2). As the FI was only measured from 21 months onwards could some of the correlations been biased “towards” very old animals. If so, would this influence the modelling?

Another good point. The specific items of the frailty index definitely change with age at different rates/points and some likely contribute more to the models at specific ages than others. Fortunately, the non-parametric nature of the random forest models is able to take this into account and does not assume linear relationships between the variables and outcomes, which is the major reason why we selected it over other model types. We now discuss this in the manuscript.

Reviewer 3

Minor comments

1. Figure 3 and 4, panels D and E -- an N of 51 mice were split for training and testing subsets, however; the panel includes roughly 100 data points -- I assume this includes the same mice at different ages, in which case, perhaps this should be explained in the legend or the ages identified as in figure 3G.

Thank you. Your assumption is correct, the datapoints represent individual frailty assessments, which were performed multiple times per mouse. This has been clarified in the legends and methods. We also have corrected an error in our reporting the N value due to the staggered timing for some of the mice on the study. We in fact had 60 mice, but this has not affected any of our analyses.

2. The Random forest model was chosen for having a lower median error than the other approaches, although this difference was not statistically significant. Therefore, would it make sense to submit all the models that are statistically similar for training, and then subsequent testing to identify which provides the strongest correlation?

Thanks for this comment – you are correct! These comparisons were obtained by bootstrapping, which is very similar to the method you suggest, where the training set is itself subdivided into training and testing sets (100 times in our analyses), and the results are averaged. We held out the true testing set for only reporting the final accuracy of the chosen model to prevent overfitting. While the differences were not statistically significant, we chose to focus on the random forest model as it is non-parametric, and able to include non-linear relationships between variables. This justification has been expanded in the methods and discussion sections of the paper.

3. The authors indicate this is the first study to, "measure frailty longitudinally in a population of naturally aging mice that were tracked until their natural deaths in order to predict health span and lifespan." Although the distinction of the goal of predicting healthspan and lifespan is acknowledged, many of these concepts were examined recently using physical frailty assessment tools (Baumann et al., Aging 2018 PubmedID: 30562163)

Thank you for pointing this out. Baumann et al 2018 used a different frailty tool and did not predict age or lifespan but they did measure frailty longitudinally and associate it with mortality. We have made the above statement clearer (we now use the term clinical frailty index, rather than frailty) and added reference of their paper to the discussion.

4. Although it is a supplement, an additional description of what is being observed in figure S1 a-f would aid readability and understanding.

We have added further description of each of these items in the figure legend for Figure S1, and referred readers to the paper's website for more detail in how to score the frailty items. The website includes a video overview and a pdf reference sheet with examples of each of the frailty index items.

Reviewer 4

General comments

1. The main claim of the paper regarding originality seems to be that it is 'the first study to measure frailty longitudinally in a population of naturally aging 209 mice that were tracked

until their natural deaths in order to predict healthspan and lifespan' (first sentence of the Discussion section). ... it should be briefly clarified the difference between this work and other works on frailty index using longitudinal data. Two such works are (Hoogendijk et al. 2017) (reference below) and (Rockwood et al. 2017).

As far as we are aware, ours is the first study to use frailty measures in mice (and in fact in humans) to predict individual outcomes, e.g. individual lifespan. There are many human studies that have shown that frailty is predictive of mortality but at the population level using risk statistics, including Hoogendijk et al. (2017). In mice, Rockwood et al (2017), did show that frailty was associated with mortality using Kaplan-Meier curves or cox regressions, but they did not predict individual outcomes. In this study we used longitudinal measures of frailty to predict individual mortality outcomes in mice. We have clarified the discussion with clearer comparisons between our work and these cited publications.

2. The set of regression techniques used seems appropriate overall, but there is not much justification for their choice. The way I see it, they represent a good diversity, varying from simple linear regression, with the advantage of easy interpretability but the disadvantage / limitation of being strongly parametric and considering only linear relationships between the predictive variables and the target variable (see also another related comment below), to more powerful random forests, with the advantage of being non-parametric and detecting non-linear relationships but the disadvantage of being more difficult to interpret. Actually, it is not practical to interpret all trees in the forest, although there are kind of indirect interpretations based on ranking the variables in decreasing order of importance in the model, as used in the paper. I think it would be good to have a short discussion about these issues in the paper.

Thank you, we have expanded consideration of the regression approaches used and justification for their choice in the revised manuscript.

3. This work uses standard versions of regression algorithms (least squared, elastic net and random forests). There are versions of these algorithms developed specifically for survival analysis (e.g. random survival forests), but I assume these survival analysis-oriented versions were not used for two reasons: (a) although predicting survival time, there is no censored data, since all animals were followed until death; and in the absence of censored data the survival problem can be cst (although not perfectly) as standard regression; (b) each run of a regression algorithm is basically considering variables at a single time point, with a few exceptions (this is discussed in more detail below). This kind of rationale could be briefly mentioned in the Methods section.

Thank you. We have expanded this justification in the methods section.

4. In any case, standard regression methods like least square assume the target variable is normally distributed, which is unlikely to be the case for survival times. By contrast, random forests, as a non-parametric method, make no such assumption. This kind of issue should be briefly discussed in the Methods section.

We have also expanded this justification in the methods section.

5. I also think the way the longitudinal information was used could be clearer. If I understand it correctly, although the data is longitudinal in nature, each run of a standard (non-longitudinal) regression technique for predicting mortality is just using the variables measured at a given point in time, the time point where the estimation of survival / mortality is done. For instance, when training a regression model for predicting mortality at age 24 months, regarding the individual frailty index items, only the predictive variables measured at age 24 are considered, predictive variables at earlier or later ages are ignored. Temporal information is considered by the regression algorithms predicting mortality only in the case of a few variables explicitly encoding temporal information, namely total percent weight change, recent percent weight change and threshold recent weight change. This issue should be clearer. (If this understanding is not correct, that would suggest an even stronger need to explain more clearly exactly how the longitudinal data was used, since other readers may get confused too.)

Thank you for highlighting this. You are correct. Each time point of frailty assessment for each mouse is treated as independent data. We now explain in more detail how we used longitudinal data and included this caveat in the limitations section of the discussion.

6. In line 385 it is said there are in total 2,460 'data points'. What exactly is a 'data point' in this context? How exactly was this total number calculated?

Thank you for pointing this out. The correct number should be 7859, which refers to the number of frailty assessments included in this study (271 total across 60 individual mice), multiplied by the number of frailty items assessed (29 items). We have corrected and clarified this in the new manuscript.

7. Several relevant related works on aging clocks are discussed, but considering that the title of the paper refers to machine learning, and considering that the state of the art in terms of predictive performance in machine learning is often deep learning (although with some limitations, see below), it seems reasonable that the discussion should be extended to mention also related work using deep learning for developing ageing biomarkers or estimating biological ageing. Such works are reviewed e.g. in: Gialluisi et al. Machine learning approaches for the estimation of biological aging: the road ahead for population studies. Frontiers in Medicine, 6, Article 146, July 2019 and Zhavoronkov et al. Deep biomarkers of aging and longevity: from research to applications. Aging 2019, 11(22), 10771-10780. Regarding the limitations of deep learning methods, normally they require a very large amount of data, which would seem to prevent their use in this current work, and they also produce models which are very hard to interpret, so I think it is justifiable not to use them in this particular work. However, the aging clocks produced by such deep learning methods should be briefly discussed as part of related work, as mentioned above.

Good points. We now discuss previous machine and deep learning approaches to aging biomarker development and cite these two papers.

8. In the experiments, FRIGHT was the best predictor of age and AFRAID the best predictor of mortality. I understand that, when training the AFRAID regression models, one of the predictive variables was chronological age. I wonder what would be the effect of replacing chronological age with the biological age estimated by the FRIGHT clock, when training the AFRAID model. Would that increase or decrease the predictive performance (on the test set) of AFRAID? That would be one approach to combine FRIGHT and AFRAID, a step in the direction of a hybrid clock as suggested in lines 283-285 of the Discussion section, and a simple experiment to do.

Great idea! We have performed this modelling and have included the results in supplementary Figure S4c-f. Surprisingly, the AFRAID models were just as predictive of lifespan when using FRIGHT age rather than chronological age. This indicates that life expectancy can be accurately predicted from frailty index items alone, without using chronological age as a variable which we now discuss.

9. In addition to reporting results in terms of r^2 and median error, it would be good to report results in terms of Mean Absolute Error (MAE) too. The median error is a good measure of central tendency, but it should be complemented by a measure which captures information about errors across all samples in the test set, like the MAE.

Thank you, we have added mean error to the manuscript text, and figures, for all testing and training sets (Figure 1d-e, 3d-e and 4d-e). For the bootstrapping data we have added a supplementary figure showing mean absolute error for each of the models (Supplementary Figure S4a-b).

10. The last paragraph of the Discussion section is discussing some limitations of this work and future research directions, overall good, but I think it is important to remind the reader here that this study has the limitation of using only data from male mice, and perhaps cite here a couple of papers showing the importance of studying variations of frailty in males vs females.

Thank you. This is a limitation that we had overlooked but now added to the discussion.

11. Line 228 mentions the clock calculators were made available online, which is good, but to allow reproducibility of the results, the full dataset (with $n = 51$, all the 30 or so items used to calculate the FI, and the survival time for each mouse) should be made available online too; this is very important.

You are correct. This has been added to our github page: <https://github.com/SinclairLab/frailty>.

Specific Comments

12. The title of Table 1 should specify that these results were computed on the test set (assuming this was the case).

Thank you. Yes, these were computed on the test set, and this has been added to the title of this table. We also adjusted the n values of this table to represent only the test dataset.

13. Lines 175 and 176: *'The AFRAID clock was also correlated with survival at individual ages (Figure 4g) with $r^2 > 0.1$ and p value < 0.05 at 24, 27, 28.5, 30 and 34.5.'* This sentence is relying purely on statistical significance ($p < 0.05$) to highlight the main correlation results, almost ignoring the magnitude of the r^2 value. Statistical significance does not imply practical significance – see e.g. the following two references for a discussion why statistical significance is over-rated in practice and the limitations of using only p -values to select the significant results to be highlighted: Amrhein, V., Greenland, S., McShane, B. (and more than 800 signatories) Retire statistical significance. *Nature*, Vol. 567, 21 March 2019, pp. 305-307 and Ronald L. Wasserstein, Allen L. Schirm & Nicole A. Lazar (2019) Moving to a World Beyond " $p < 0.05$ ", *The American Statistician*, 73:sup1, 1-19, DOI: 10.1080/00031305.2019.1583913. Although the issue of which r^2 value is high enough to be considered a strong or at least reasonable degree of correlation is obviously subjective, it seems reasonable to say that an r^2 of around 0.2 is not high enough to be relevant in practice, regardless of p -value. Out of the 5 age values mentioned above, the r^2 for 27 and 28.5 months are 0.218 and 0.191, much lower than the r^2 for the other 3 above ages. I would suggest to say that there are 3 ages with $r^2 > 0.3$ and statistically significant results, mentioning explicitly the $r^2 = 0.653$ for age 34.5 and $r^2 = 0.447$ for age 24, rather than reporting all the above 5 age values with r^2 having $p < 0.05$.

This has been changed as suggested.

14. Lines 245-247 state that the AFRAID clock predicted mortality with a median error of 46 days across multiple ages. I understand this is the median error on the test set. In figure 4(g), the median error of AFRAID on the test set is 1.7 months. Assuming a month = 30 days, 1.7 month gives a median error of 51 days, not 46 days as stated in line 247.

Corrected.

15. Line 99 mentions median error of 1.8 months, r -squared 0.642, and $p = 7.3e 20$. The first two of these three values are the same as in Figure 1(e), and indeed the whole sentence seems to be about performance on the testing set, but the third above value, $p = 7.3e 20$, was wrongly taken from Figure 1(d), referring to performance on the training set. In Figure 1(e), the value is $p = 3.4-38$.

Corrected.

16. Lines 322-323 (Methods – Animals), it is stated that in the rare cases where mice were euthanized, the date of euthanasia was taken as the best estimate of death. The phrase 'In these rare cases' is vague. To be more precise, please give the exact number of such rare cases.

We have updated the methods with the exact criteria for euthanasia and highlighted that only 4 out of 60 mice were euthanized.

17. Lines 394-395 (Methods – Random Forest), it is stated that ‘the minimum number of samples required for a branch split was limited...’, which is vague. To be precise, please give the exact value of the threshold used as minimum number of samples for a branch split. Actually, all settings of hyper-parameters of the RF algorithm have to be given, so please give also the used value of the mtry parameter (the number of randomly sampled variables used as candidates for branch split at each tree node), as well the settings of any other hyper-parameters of the RF algorithm (and for the other regression algorithms used too). Even if the authors are using default parameter settings, it is worth mentioning explicitly their exact values, because default values in the software implementing an algorithm can change with time, and so citing the precise hyper-parameter settings makes the paper more self-contained.

Details of the hyperparameters have been added to the methods section for the elastic net and random forest models, including the specific min_leaf_sample thresholds. Mtry, which is called max_features in the Python sklearn package, was set to “auto” which includes all the features. We have also made the code of our models available on github (<https://github.com/SinclairLab/frailty>) where all of the features can be viewed with the model.get_params() function.

Minor comments

18. Lines 99, 139, 616: replace r-squared by r^2 (using superscript, of course, which does not appear in this plain text web form)

Corrected in all instances.

19. Line 174, replace 1.1-26 by 1.1e-26

Corrected.

20. Line 305, replace expectency by expectancy

Corrected.

Peer Review File

Comments second round –

Reviewer #2 (Remarks to the Author):

The revised version of the manuscript addressed all questions and comments of my initial review. Making the analysis publicly available added a lot of value.

Reviewer #3 (Remarks to the Author):

The authors have strengthened the manuscript. One minor concern remains. Please include cage conditions, including whether singly or group house, and the type of caging in the description of the experiment.

Reviewer #4 (Remarks to the Author):

The authors have addressed well my comments in the first round of reviews, and at this point I have just a few minor suggestions or corrections, as follows.

In the Section Method, subsection Modelling and Statistics, lines 401-402:

'All models were assessed with bootstrapping with replacement, repeated 100 times.'

This could be made clearer by adding here some text which is a small variation of the text used by the authors in their response to a comment by another reviewer, where the authors wrote:

'These comparisons were obtained by bootstrapping, ..., where the training set is itself subdivided into training and testing sets (100 times in our analyses), and the results are averaged. We held out the true testing set for only reporting the final accuracy of the chosen model to prevent overfitting.'

I just suggest a small variation in this text, considering that in this particular phrase:

'the training set is itself subdivided into training and testing sets',

it is best to avoid calling those data subsets by the same names used for the original training and testing sets, to avoid confusion. You can say the training set is divided into sub-training and validation sets, for example.

So, my suggestion is to expand the text in lines 401-402:

'All models were assessed with bootstrapping with replacement, repeated 100 times'

into something like this:

'All models were assessed with bootstrapping with replacement, repeated 100 times. In each of those 100 iterations, the training set is divided into sub-training and validation sets, and the results on the validation sets are averaged over the 100 iterations. We held out the testing set for only reporting the final accuracy of the chosen model to prevent overfitting.'

A few typo corrections:

Line 80, replace 'predicitve' by 'predictive'

Line 455, replace 'max_meatures=auto' by 'max_features=auto'

Line 786, replace 'two decision tree' by 'two decision trees'

We thank the reviewers for their positive words and suggestions. We have addressed the comments of the reviewers as outlined below and highlighted in the final manuscript in red text.

Reviewer 3

1. Please include cage conditions, including whether singly or group house, and the type of caging in the description of the experiment

Thank you, we have included this in the manuscript as below:

Aged males C57BL/6Nia mice were ordered... and housed at Harvard Medical School in ventilated caging with a 12:12 light cycle, at 71°F with 45-50% humidity. Mice were group housed (3-4 mice per cage) at the start of the experiment, although over the period of the experiment mice died and mice were left singly housed.

Reviewer 4

1. In the Section Method, subsection Modelling and Statistics, lines 401-402: 'All models were assessed with bootstrapping with replacement, repeated 100 times.'... I suggest a small variation in this text, considering that in this particular phrase: 'the training set is itself subdivided into training and testing sets', it is best to avoid calling those data subsets by the same names used for the original training and testing sets, to avoid confusion. You can say the training set is divided into sub-training and validation sets, for example. So, my suggestion is to expand the text in lines 401-402: 'All models were assessed with bootstrapping with replacement, repeated 100 times' into something like this: 'All models were assessed with bootstrapping with replacement, repeated 100 times. In each of those 100 iterations, the training set is divided into sub-training and validation sets, and the results on the validation sets are averaged over the 100 iterations. We held out the testing set for only reporting the final accuracy of the chosen model to prevent overfitting.'

Thank you for this suggestion. We have made this change, you are right – it is much clearer now.

2. A few typo corrections:

Line 80, replace 'predicitve' by 'predictive'

Line 455, replace 'max_meatures=auto' by 'max_features=auto'

Line 786, replace 'two decision tree' by 'two decision trees'

Thank you! These have been corrected.